# A CSI Fingerprint Method for Indoor Pseudolite Positioning Based on RT-ANN

**Yaning Li** [1,2,3,*], **Hongsheng Li** [1,*], **Baoguo Yu** [2,3] **and Jun Li** [2,3]

1   School of Instrument Science and Engineering, Southeast University, Nanjing 210096, China
2   State Key Laboratory of Satellite Navigation System and Equipment Technology, Shijiazhuang 050081, China; yubg@sina.cn (B.Y.); cetclj@126.com (J.L.)
3   The 54th Research Institute of China Electronics Technology Group Corporation, Shijiazhuang 050081, China
*   Correspondence: 15631149037@163.com (Y.L.); hsli@seu.edu.cn (H.L.); Tel.: +86-156-3114-9037 (Y.L.)

**Abstract:** At present, the interaction mechanism between the complex indoor environment and pseudolite signals has not been fundamentally resolved, and the stability, continuity, and accuracy of indoor positioning are still technical bottlenecks. In view of the shortcomings of the existing indoor fingerprint positioning methods, this paper proposes a hybrid CSI fingerprint method for indoor pseudolite positioning based on Ray Tracing and artificial neural network (RT-ANN), which combines the advantages of actual acquisition, deterministic simulation, and artificial neural network, and adds the simulation CSI feature parameters generated by modeling and simulation to the input of the neural network, extending the sample features of the neural network input dataset. Taking an airport environment as an example, it is proved that the hybrid method can improve the positioning accuracy in the area where the fingerprints have been collected, the positioning error is reduced by 54.7% compared with the traditional fingerprint positioning method. It is also proved that preliminary positioning can be completed in the area without fingerprint collection.

**Keywords:** CSI fingerprint; ANN; pseudolite; ray tracing; indoor positioning

## 1. Introduction

Compared with the outdoor open free space, the indoor channel environment and spatial topology are more complex, and the GNSS space signal service cannot be effectively covered in the indoor environment due to the attenuation of building occlusion. A pseudolite (PL) is a ground navigation transmitter that can transmit GNSS-compatible signals. In an indoor environment, pseudolites transmit the signals similar to the space satellite signals to the user terminal through the transmitting antenna. For location prediction, indoor and outdoor continuous positioning services can be provided through software upgrades without changing the existing hardware of smart terminals on the market [1–3].

However, in the indoor environment, the pseudolite signal will produce propagation effects such as path loss and multipath fading under the influence of multipath propagation and shadow shading. The mature theoretical methods of traditional GNSS positioning are not fully applicable for indoor pseudolite positioning. The well-known distance-based localization techniques, such as Time of Arrival (TOA), Time Difference of Arrival (TDOA), and Angle of Arrival (AOA) [4], do not work well in complex indoor environments. Fingerprint-based localization methods are not limited by line-of-sight (LOS), and have the advantages of low cost, high accuracy, and good stability, so they are widely used. The pros and cons of different indoor localization technologies such as AoA, ToF, RTOF, and RSS are discussed in reference [5], and several common fingerprint matching methods are also introduced in this article.

Fingerprint localization has been extensively studied by researchers. An indoor localization approach is presented in reference [6], which utilizes the magnetic data from smartphone magnetic sensors to localize a pedestrian. The authors train a convolutional

neural network to recognize the indoor scene, and build a database of magnetic field patterns to lower the device dependence. Reference [7] proposes a method using the intersection area of AP to calculate the user's position, which alleviates the influence of human body loss on the positioning accuracy and reduces the device's dependence on the fingerprint database. Reference [8] proposes an indoor fingerprint positioning sensor fusion framework that combines Wi-Fi RSSI signals, smartphone sensors, and the PDR algorithm, which shows reasonable localization accuracy with fewer IMU sensor errors. The authors of [9] propose an algorithm that utilizes high-level features extracted by deep learning, extreme learning machines, and autoencoders to improve localization performance in feature extraction and classification, while increasing the amount of training data to improve localization performance. The authors of [10] use the K-Nearest Neighbor (KNN) algorithm to study the accuracy of wireless fingerprint localization and compare its performance with other fingerprint localization algorithms. Reference [11] proposes an indoor positioning algorithm that combines fingerprint positioning and dynamic prediction. The proposed algorithm can alleviate the influence of received signal strength (RSS) fluctuations and has better positioning accuracy and stability.

In addition to the above-mentioned RSS, the data form currently obtained based on indoor positioning fingerprints also includes channel state information (CSI). RSS represents the signal strength value superimposed at the receiving point. When it is used as the only fingerprint feature, much useful information such as phase are ignored, so that the propagation characteristics of the signal in the channel are not well-reflected. RSS can only achieve ideal results in some simple environments, and CSI is more stable than RSS in traditional indoor environments. CSI contains both amplitude and phase information, and can provide richer frequency domain information than single-valued RSS. In conclusion, CSI has better stability and location sensitivity than RSS, which provides a new idea for achieving a more robust and accurate indoor positioning effect.

Reference [12] proposes an indoor fingerprint positioning method based on carrier phase difference, which uses GPS/BDS-compatible satellite signals transmitted by pseudolite base stations, compared with RSS fingerprints, the positioning accuracy is greatly improved. Reference [13] proposes a localization algorithm that combines RSS and CSI. The algorithm selects high-correlation RSS and CSI based on deep learning to build a fingerprint library, which improves the localization accuracy. Reference [14] proposes a localization method using autoregressive (AR) modeling entropy of CSI magnitude as a location fingerprint, which not only has the advantage of simple structure, but also retains location-specific statistical channel information. The experimental results show that the method significantly improves the localization performance. Reference [15] constructs a fingerprint database based on CSI values, further reduces its dimensionality by using a multi-dimensional scaling algorithm, and then uses KNN to obtain the estimated target location. Reference [16] proposes an EKF ranging scheme based on CSI and RSSI, which solves the ranging accuracy problem under high-load APs. The experimental results show that the method effectively improves the accuracy in an indoor environment.

Two methods are widely used to obtain CSI, one is a statistical method based on actual channel measurement data, and the other is a mathematical method based on the accurate calculation of electromagnetic data. The first method is based on actual measurement data and has poor adaptability to different environmental characteristics, while the second method relies on ideal simplifications and assumptions about the environment, which brings an irreparable deviation from the actual situation. In general, deterministic methods are more accurate than statistical methods because they are not affected by a single acquisition error. But the deterministic model requires detailed prior environment information (3D map or scene model), resulting in high computational complexity.

Fingerprint-based indoor positioning technology aims to find the functional relationship between eigenvalues and positions through known CSI information, and predict indoor positions online through matching technology. In a broad sense, neural networks can be introduced wherever there is a prediction, classification, or control problem [17,18].

In reference [19], a deep neural network is used for indoor positioning based on the magnetic field. The neural network plays an important role in mitigating the influence of equipment heterogeneity and improving indoor positioning accuracy. As an efficient and accurate learning matching algorithm, neural networks can process fingerprint information to achieve more accurate positioning effects. This is due to two important advantages: the first is that the neural network is composed of interconnected neurons and can be used to model highly distributed and highly parallel problems; the second is that the neural network can learn functional relationships on the basis of the problem to be solved. At present, researchers usually focus on the improvement of network architectures and lateral comparisons of various machine learning methods [20–23], and there are relatively few studies on the diversity and multidimensionality of neural network inputs.

In summary, the CSI fingerprint localization method based on the neural network can more accurately describe indoor environment characteristics and realize real-time localization. However, most of the data sets used for training are actually collected data, and there are two major limitations as follows: (1) Location problem of unknown environment. The positioning technology using the actual collected fingerprints based on a controlled experimental environment, the positioning service will not be able to be provided in the uncollected area; (2) In order to achieve full coverage of the indoor environment, multiple APs are needed as beacons, and the positioning accuracy is related to the density of test points. High-precision location service in large indoor venues means huge testing workload, and is greatly affected by a single acquisition error. Some researchers use the propagation model to calculate the virtual APs to reduce the deployment of the actual Aps [24–27]. However, errors caused by simplifying different spatial propagation parameters are still inevitable. On the other hand, the fingerprint feature collection at the test point needs to actually measure the information of the surrounding APs, and the workload cannot be reduced accordingly. Some researchers use the existing sparse fingerprint database data to generate a dense fingerprint database by interpolation algorithm, but it may fail in some large scene areas (such as shopping malls, gymnasiums, etc.) since the signal features of these sparse reference points cannot represent the entire positioning area. Moreover, the running time of the algorithm will increase rapidly with the increase of the number of reference points, resulting in that the real-time performance of the positioning cannot be guaranteed.

This paper proposes an indoor pseudolite simulated CSI fingerprint positioning method that combines measured data, deterministic calculation data, and an artificial neural network. The channel characteristic parameters obtained by the deterministic calculation method are used as the training set of the neural network together with the actual measurement data. The extended input set improves the positioning accuracy in the area where the fingerprints have been actually collected, and can also complete the preliminary positioning in the area where the fingerprints are not actually collected. The main work of this paper is summarized as follows:

1. Selecting position-specific CSI parameters, and deriving the conversion relationship between the simulation parameters of the ray tracing method and the actual CSI parameters;

2. An indoor pseudolite CSI fingerprint positioning method combining measured data, deterministic calculation data, and artificial neural network were proposed. On the basis of the original fingerprints input, the simulation feature parameters were added, expanding the sample features of the neural network input dataset.

3. Build an indoor test environment in the arrival hall of an airport, and verify the positioning performance of the proposed algorithm in a large indoor environment.

The rest of the paper is organized in the following manner. Section 2 presents an overview of the indoor pseudolite fingerprint positioning method from the aspects of fingerprint feature selection and fingerprint positioning principle. Section 3 introduces the basic principle and calculation formula of the ray tracing method. Section 4 describes the proposed approach while Section 5 details the experiment setup and analyzes the results. Finally, conclusions are given in Section 6.

## 2. Indoor Pseudolite Fingerprint Positioning Method

### 2.1. Selection of Feature Fingerprints for Pseudolite Observations

The original observation information output by the pseudolite receiver includes the pseudorange, carrier phase, integral Doppler, carrier-to-noise ratio, and other observational values of each epoch [28]. The location specificity and stability of each observation are different and need to be analyzed and selected.

(1) Pseudorange observations

Pseudorange observation is the most basic observation in GNSS positioning, and is a necessary condition for absolute single-point positioning. In complex indoor environments, it can directly reflect the distance between the pseudolite transmitting antenna and the receiving antenna, but the pseudorange measurement value has low precision and poor stability due to the influence of multipath propagation.

(2) Carrier Phase Observations

The carrier phase observation is another basic observation output by the receiver, which represents the difference between the phase of the carrier signal reproduced by the receiver and the phase of the carrier signal received by the receiver. The carrier phase difference between channels is more stable than the pseudorange difference, and is basically a constant value, that is, the distance difference between the pseudolite antennas and the receiving antenna. Carrier phase difference can eliminate the effects of receiver clock differences, but cannot eliminate the effects of space-specific multipath, so it can be added to the signature database as an information source.

(3) Carrier-to-noise ratio Observations

The carrier-to-noise ratio ($C/N_0$) observation is the ratio between the received carrier signal strength and the local noise, describing the quality of the signal received by the receiver. Although the observed value of $C/N_0$ has a certain relationship with the signal processing capability and signal bandwidth of the receiver itself, $C/N_0$ can clearly reflect the distance or occlusion between the receiver and the transmitter when the same receiver is used to receive the same system signal.

Since no dynamic measurements are involved, Doppler observations are not concerned. Considering the convenience of measurement and the stability of data, we use the simulating channel parameters TOA and RSS, corresponding to the carrier phase and $C/N_0$ in the measured CSI data as eigenvalues for position prediction. Through actual measurements in an indoor environment, the data stability and location specificity are shown in Figure 1. It can be seen that both have good spatial resolution and stability, clearly indicating that these two parameters can be used as the input of the neural network.

### 2.2. Positioning Principle

Pseudolite observational feature fingerprint matching positioning technology is based on the correlation between the original observational information output by the receiver and the indoor physical location. Specifically, positioning is performed by using different signal responses reflected by multi-dimensional observations at different indoor locations. Figure 2 is a schematic diagram of the principle of pseudolite fingerprint positioning.

The positioning process is mainly divided into two phases: offline data acquisition and online position calculation. In the offline collection phase, sample collection is performed at the reference point, the feature values of each sample parameter are extracted, and the function mapping relationship between them and the physical reference position is learned to construct a feature fingerprint database. In the online phase, the real-time observations of the receiver are input into the feature fingerprint database, and the positioning result is obtained through matching algorithm [29–31].

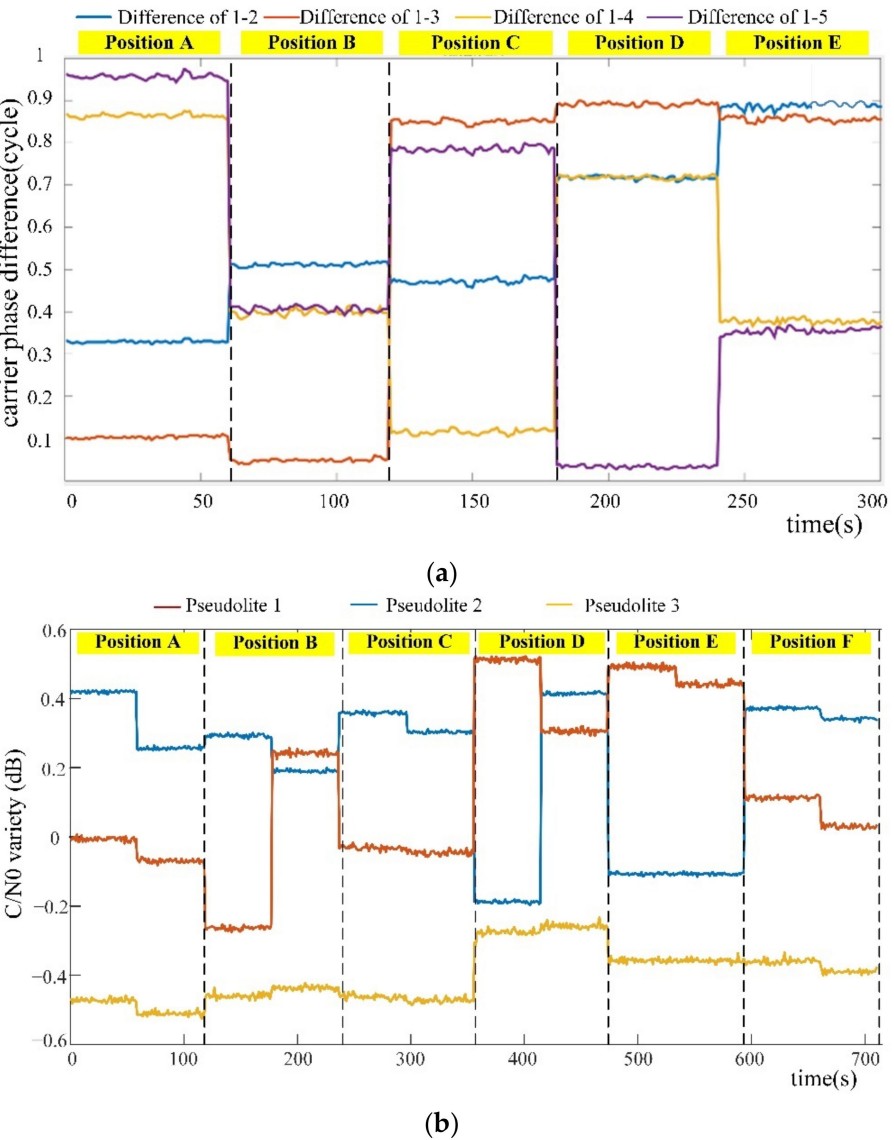

**Figure 1.** Location specificity and stability of carrier phase difference and $C/N_0$. (**a**) Location specificity and stability of carrier phase difference. (**b**) Location specificity and stability of $C/N_0$.

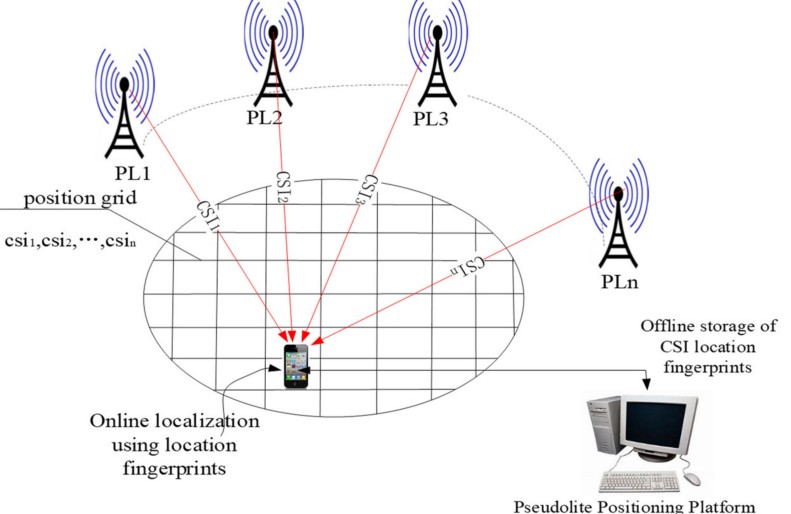

**Figure 2.** The principle of pseudolite fingerprint positioning.

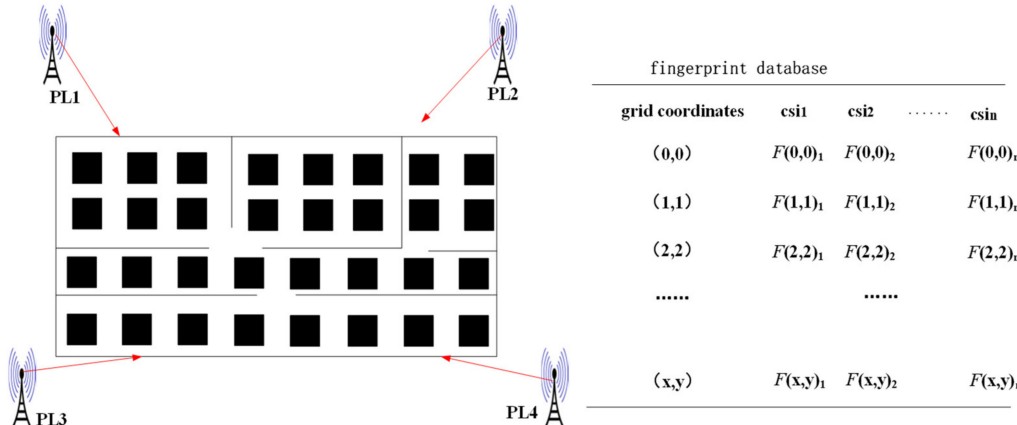

**Figure 3.** Establishment of location fingerprint database.

(1) Offline phase

The correspondence between location and fingerprint is established in the offline phase, as shown in Figure 3. The space area is gridded, and pseudolites are deployed around the service area to ensure that all user receivers can receive the pseudolite signal. At each grid point, the channel observations from each pseudolite receiver are obtained through data sampling over a period of time. The grid point coordinates and the corresponding CSI fingerprints form a fingerprint feature database. This process is sometimes called the labeling stage, and this fingerprint database is sometimes called a signal map.

(2) Online Phase

In the online positioning phase, we compare and calculate the CSI eigenvalues of the to-be-located point with the eigenvalues in the fingerprint database, and obtain a set of closest eigenvalue parameters in the fingerprint database to obtain the corresponding position coordinates, thereby completing the positioning of the receiver. The matching method used in this paper is the nearest neighbor selection method:

First calculate the Euclidean distance between the feature value group $(F_1, F_2, \ldots, F_n)$ of the point to be located and the reference feature value group in the fingerprint database, the details are as follows:

$$L_i = \sqrt{\sum_{j=1}^{n} \left| F_j - F_{ij} \right|^2} \tag{1}$$

where $L_i$ is the Euclidean distance between the point to be located and the $i$th reference point in the fingerprint database. $F_j$ is the eigenvalue of the $j$th dimension in the eigenvalue group of the point to be located, $F_{ij}$ is the eigenvalue of the $j$th dimension of the $i$th reference point in the eigenvalue fingerprint database. $i = 1, 2, \ldots, m, \ j = 1, 2, \ldots, n$.

Then select $k(k \geq 2)$ reference points from near to far in $L_i$, the position coordinates of the point to be located can be estimated from these k reference points:

$$(\hat{x}, \hat{y}) = \frac{1}{k} \sum_{i=1}^{k} (x_i, y_i) \tag{2}$$

where $(\hat{x}, \hat{y})$ is the estimated coordinate of the point to be located; $(x_i, y_i)$ is the coordinate of the $i$th point in the $k$ nearest reference points.

The weighted k-nearest neighbor (WKNN) classification algorithm assigns a weight to each nearest neighbor reference point coordinate, the weight value is obtained by Gaussian transformation from the distance $L_i$ between the to-be-located point and the nearest reference point:

$$W_{ki} = ae^{\frac{(L_{ki} - L_{imin})^2}{2b^2}} \tag{3}$$

where $W_{ki}$ is the weight of the $i$th point in the k nearest reference points, $L_{ki}$ is the distance from the to-be-located point to the $i$th point in the k nearest reference points, $L_{imin}$ is the minimum distance from the to-be-located point to the $i$th point in the $k$ nearest neighbor reference points. $a$ is the maximum weight value, which is usually 1; $b$ is the half-peak width, which can be adjusted according to the value of $k$.

The coordinates of each adjacent reference point are combined with the corresponding weight, and the coordinates of the point to be located can be obtained by the WKNN algorithm:

$$(x, y) = \frac{\sum_{i=1}^{k} W_{ki} \times (x_i, y_i)}{\sum_{i=1}^{k} W_{ki}} \tag{4}$$

## 3. Ray Tracing

### 3.1. Algorithm Principles

Ray Tracing (RT) is a method of simulating high-frequency electromagnetic waves in the research environment into light waves. Combined with the consistent diffraction theory, it is widely used to study the propagation of high-frequency electromagnetic waves [32–35]. On the premise that the signal frequency belongs to the high frequency band, the signal can be regarded as a ray propagating along a straight line in space and interface. The pseudolite signal adopts the L frequency band, which belongs to the ultra-high frequency band, so it conforms to the theoretical premise of ray tracing.

Specifically, the pseudolite is abstracted into a point source that emits electromagnetic rays in all directions, and electromagnetic parameters are calculated for each propagating ray by electromagnetic calculation methods, as shown in Figure 4. After the ray reaches the receiving end R, the detailed information such as the arrival amplitude, propagation delay relative to the first path, arrival phase, and arrival angle of each ray is obtained, then the vectors of all arriving rays are superimposed to obtain the simulation data of indoor pseudolite signals at the receiving point.

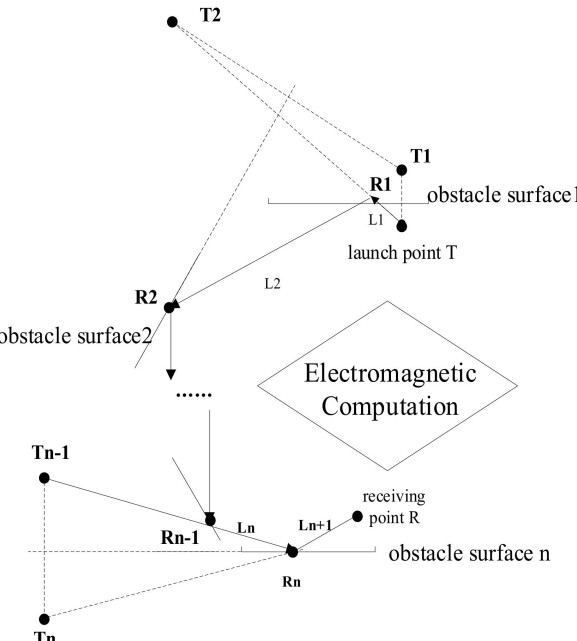

**Figure 4.** Mirror ray propagation process.

Ray tracing is a deterministic calculation method using Maxwell's equations, all electromagnetic rays are traced back to the emission source according to the imaging theory, and the computational efficiency of the model is highly dependent on the complexity of the environment. The direct transmission of rays is also called LOS. When the rays are reflected on the surface of the object, the direction of the generated signal rays is determined by

the reflection and refraction characteristics of the surface material [36], and the energy is calculated by the Fresnel equation. Similarly, when rays penetrate an object (transmission) or diffract and scatter at the edges of the object, the Fresnel equations are combined with imaging theory to determine the energy and direction of the resulting rays.

*3.2. Typical Characteristic Parameters*

(1)   Received power [37–40]

The total received power at the receiving point is:

$$P_R = \sum_{i=1}^{N_P} P_i \tag{5}$$

where $N_p$ is the number of paths and $P_i$ is the time-averaged power of the $i$th path.

$$P_i = \frac{\lambda^2 \beta}{8\pi\eta_0} \left| E_{\theta,i} g_\theta(\theta_i, \phi_i) + E_{\phi,i} g_\phi(\theta_i, \phi_i) \right|^2 \tag{6}$$

where $\lambda$ is the wavelength and $\eta_0$ is the impedance of free space (377 $\Omega$), $E_{\theta,i}$ and $E_{\phi,i}$ are the $\theta$ and $\phi$ electric field components of the $i$th path at the receiving point, $\theta_i$ and $\phi_i$ indicates the direction of arrival.

The direction of arrival of the signal at the receiving point is given by:

$$g_\theta(\theta, \phi) = \sqrt{|G_\theta(\theta, \phi)|} e^{j\varphi_\theta} \tag{7}$$

where $G_\theta$ is the $\theta$ component of the receiving antenna gain and $\varphi_\theta$ is the relative phase of the $\theta$ component of the electric field in the far region. $\beta$ is the overlapping part of the frequency band of the transmitted signal $S_T(f)$ and the frequency band that the receiver $S_R(f)$ can receive:

$$\beta = \frac{\int_{f_T-(B_T/2)}^{f_T+(B_T/2)} S_T(f) S_R(f) df}{\int_{f_T-(B_T/2)}^{f_T+(B_T/2)} S_T(f) df} \tag{8}$$

where $f_T$ and $B_T$ are the center frequency and bandwidth of the transmitted waveform, respectively. At present, the narrowband waveforms are assumed to be flatly distributed:

$$S(f) = \begin{cases} 1 & f_0 - \frac{B}{2} < f < f_0 + \frac{B}{2} \\ 0 & \text{others} \end{cases} \tag{9}$$

where $f_0$ is the center frequency and $B$ is the bandwidth.

The total received power is:

$$P_R = \frac{\lambda^2 \beta}{8\pi\eta_0} \left| \sum_{i=1}^{N_p} \left[ E_{\theta,i} g_\theta(\theta_i, \phi_i) + E_{\phi,i} g_\phi(\theta_i, \phi_i) \right] \right|^2 \tag{10}$$

(2)   TOA

The arrival time of each propagation path is:

$$t_i = \frac{L_i}{c} \tag{11}$$

where $L_i$ is the total length of the $i$th electromagnetic ray path, and $c$ is the speed of light.

## 4. Fingerprint Localization Method Based on RT-ANN

The neural network architecture for training fingerprint positioning in the existing public literature is generally shown in Figure 5, and the data of the input layer comes from the CSI information measured in the environment [41–45]. This approach leads to many

inherent errors and inconveniences: (1) There are many obstacles in the indoor environment. The material and thickness of walls, floors, doors and windows have a great impact on the propagation of indoor pseudolite signals, and the process of environmental characterization is more complicated; (2) There are inherent errors that cannot be eliminated by relying only on measured data for model training, such as personal factors of testers, errors of the receiver itself, errors of environmental dynamic changes, etc. (3) Only relying on the actual measurement to establish a fingerprint database, the positioning coverage and positioning accuracy are limited by the collection workload.

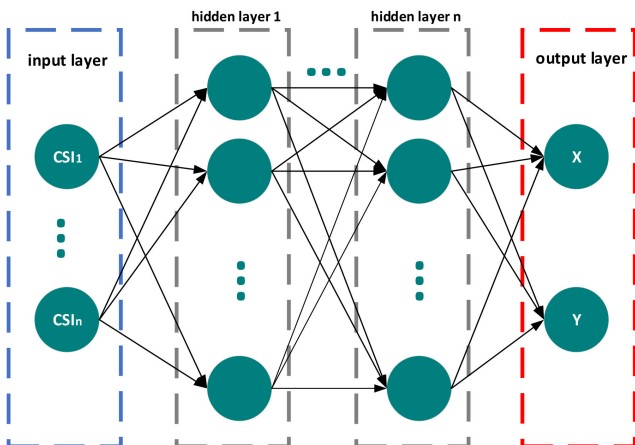

**Figure 5.** ANN network structure.

In view of the feasibility of the artificial neural network for channel modeling and the shortcomings of statistical fingerprint input data, a hybrid algorithm of artificial neural network and ray tracing (RT-ANN) is designed in this paper, which uses the characteristic parameters generated by deterministic modeling to improve the original artificial neural network, that is, the location of the receiving point of an indoor scene is jointly trained and predicted by the lower density measurement data set and the higher density simulation data set of the area.

The $C/N_0$ in the measured data corresponds to the received power in the simulation parameters. The received power at the receiving point is obtained through ray tracing deterministic simulation and then converted to $C/N_0$. The conversion process is as follows.

The signal-to-noise ratio at the receiving point is expressed as:

$$SNR = \frac{S}{N_0 B_n} \tag{12}$$

$$SNR(dB) = \frac{C}{N_0(dB \cdot Hz)} - B_n(dB) \tag{13}$$

where $B_n$ is the filter bandwidth of the receiver, $S$ is the received signal power, $N_0$ is the noise power spectral density.

The conversion relationship between $C/N_0$ and received power is as follows:

$$\frac{C}{N_0} = 10lg\left(\frac{S}{N_0 B_n}\right) + B_n(dB) \tag{14}$$

As for carrier phase positioning, the distance between satellite S and receiver R can be described by:

$$\rho = \lambda(\varphi_S - \varphi_R) \tag{15}$$

where $\rho = c \cdot TOA$, $c$ is the speed of light, $\lambda$ is the carrier wavelength, $\varphi$ is the carrier phase. It is more accurate to use the carrier phase difference considering the influence of the clock difference:

$$\Delta\rho = \lambda \cdot \triangle (\varphi_S - \varphi_R) = c \cdot TDOA \tag{16}$$

Since the actually measured carrier phase has the strongest correlation signal after demodulation, the simulated carrier phase also corresponds to the path with the strongest power; thus, Equation (16) can be written as:

$$\triangle (\varphi_S - \varphi_R) = \frac{c}{\lambda} \cdot (TOA_{Smax1} - TOA_{Smax2}) \tag{17}$$

The proposed algorithm framework and process are shown in Figure 6.

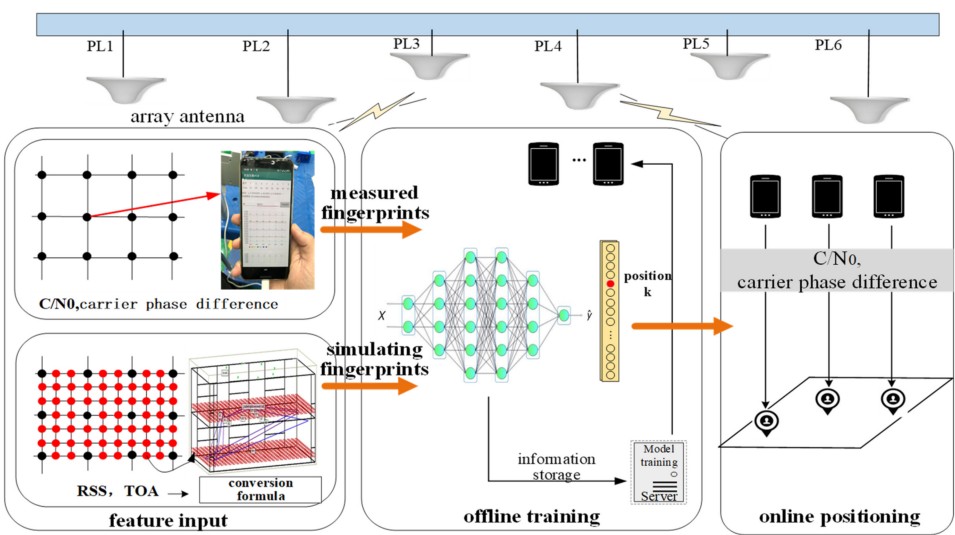

**Figure 6.** RT-ANN network structure.

For accurate representation of more complex environments, the deterministic simulation parameters of the training data set are added. Since the deterministic modeling process comprehensively considers information such as the layout and material of the building, the environmental characteristics can be described more accurately. On this basis, it is considered to increase the number of hidden layers of the original neural network, so as to attain multi-level abstraction of input features, especially a fully connected Multilayer Perceptron (MLP) between different layers can obtain a more robust channel model. In addition, the forward modeling process based on deterministic computing can just eliminate random factors such as personal factors of testers, errors of the receiver itself, and errors of environmental dynamic changes. The specific implementation steps are described as follows:

(1) CSI fingerprints collection is performed in the actual environment to obtain the measured values of carrier phase difference and $C/N_0$ at relatively sparse test points;

(2) Modeling the environment based on a 3D map or dimensional material information of the actual environment, and then simulated CSI information such as signal arrival power and arrival delay at relatively dense test points can be calculated by ray tracing. The density of test points can be flexibly set according to the required accuracy.

(3) After the simulation data is converted by Equations (12)–(17), it is used as the input of the neural network together with the measured data.

(4) The following Algorithms 1 is used for offline model training.

The essence of an artificial neural network is to learn by adapting and modifying the weights of internal connections based on the input and desired output. During the learning process, the network can adjust the input parameters by choosing weights and biases with the goal of minimizing the loss function. Different from the past, the training data of this

network is not only the measurement data, but also the simulation data calculated by ray tracing, which expands the sample characteristics of the dataset.

The input layer contains two input eigenvectors, the corresponding measurement parameters are $C/N_0$ and carrier phase, and the corresponding simulation parameters are RSS and TOA. The neural network has four hidden layers, the first hidden layer contains 30 hidden neurons, the second layer contains 20 hidden neurons, the third layer contains 10 hidden neurons, the fourth layer contains 8 hidden neurons, the output layer contains two neurons, which are the coordinates $(x, y)$ of the positioning position. During training, the algorithm's accuracy threshold and validation checks have an impact on the training process. The training process terminates once the prediction accuracy is satisfied or the error metric of the test set does not decrease within a certain number of consecutive iterations. The model training process is shown in Algorithm 1, where Reconstruction loss and Classification loss and Kullback-Leibler loss are the loss functions.

---

**Algorithm 1:** Model Training

---

Input: Pseudolite Observation dataset: $X_1 = \left[ X_1^{(1)}, X_1^{(2)}, \ldots, X_1^{(n)} \right]$, Ray Tracing simulation
  dataset $X_2 = \left[ X_2^{(1)}, X_2^{(2)}, \ldots, X_2^{(n)} \right]$, $n$ is the number of pseudolites.
  Location label: $y$.
Output: Representation: $z$ and parameter: $\phi$; $\theta$, Classification model: $Model_{classifier}\{(z, y)\}$
  1: Initialization Parameters: Number of neurons for all layers;
    The number of iterations (epochs);
  2: while $\{\phi, \theta\}$ not converged do
  3: $D \leftarrow getMinibatch()$
  4: $\mu_\theta, \theta_\theta \leftarrow x, y$;
  5: Sampling $\varepsilon$;
  6: Sampling from the posterior $z \leftarrow q_\phi(z|x, y)$ using the flowing
    Reparameterization trick: $z = \mu_\theta + \sigma_\theta \cdot \varepsilon$;
  7: Calculate the gradient of the variational lower bound $L$ (Reconstruction loss and
    Classification loss and Kullback-Leibler loss);
  8: Minimize $L$;
  9: end while
  10: while Classification model Training do
  11: Fit $\forall \{x, y\} \in D$ train Classifier $Model_{classifier}\{(z, y)\}$
  12: end while

---

## 5. Result

### 5.1. Environment Modeling

The actual measurement and simulation experiment area is the arrival hall of an airport. The scene description is shown in Table 1. Six pseudolites are arranged on the roof in an equally divided circle. There are 6 load-bearing columns on the central axis of the hall, and their cross-section is a square with a side length of 0.5 m, the material of the wall load-bearing column is concrete, the ground material is ceramic tile, the pseudolite transceiver antenna is an an omnidirectional antenna, and the store compartment has been simplified. There are two experimental areas. Text area 1 is an area where the fingerprints have been collected. In this area, manual CSI fingerprints collection is performed first, and then deterministic modeling and simulation are carried out to expand the fingerprint dataset, there are 400 sampling points with an interval of 0.25 m; Text area 2 is an area without fingerprints collection, and only the simulation fingerprint data is used to train and predict the position, there are 100 simulated sampling points with an interval of 0.5 m. The experimental environment and method are shown in Figure 7.

**Table 1.** Simulation calculation parameters for airport model.

| Parameter | Text Area with Fingerprints Collection | Text Area without Fingerprints Collection |
|---|---|---|
| scene model size | 100 m × 20 m × 4.5 m | 100 m × 20 m × 4.5 m |
| bearing pillar size | 0.5 m × 0.5 m × 4.5 m | 0.5 m × 0.5 m × 4.5 m |
| wall material/permittivity/conductivity | concrete/5/0.0015 | concrete/5/0.0015 |
| floor material/permittivity/conductivity | tile marble/6/$10^{-8}$ | tile marble/6/$10^{-8}$ |
| signal frequency | 1561.098 MHz | 1561.098 MHz |
| antenna type | omnidirectional | omnidirectional |
| receiving area size | 5 m × 5 m | 5 m × 5 m |
| receiving point interval | 0.25 m | 0.5 m |
| transmitting power | −70 dBm | −70 dBm |
| number of reflections/transmissions/diffractions | 3/1/1 | 3/1/1 |
| ray interval | 0.25 degree | 0.25 degree |

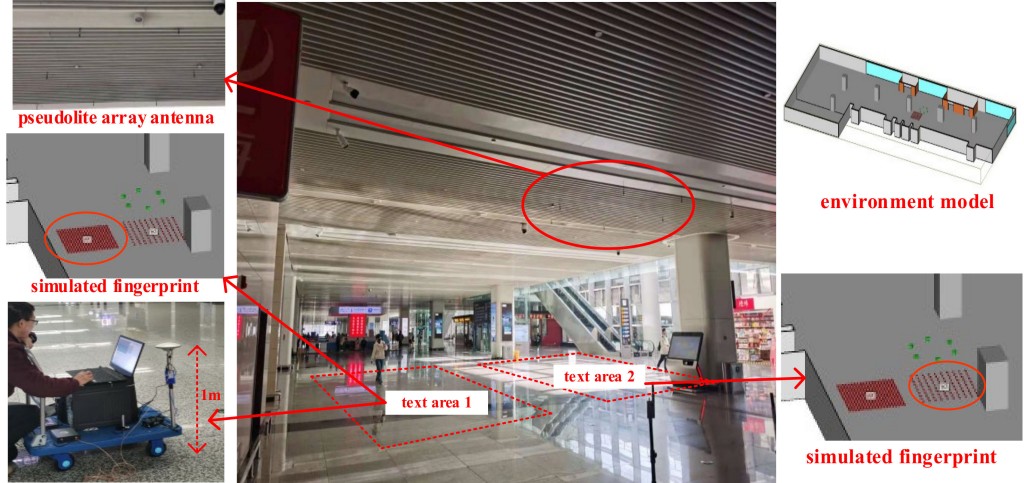

**Figure 7.** Experimental environment and method.

## 5.2. Fingerprint Data Generation

In the actual test, 25 position reference points with an interval of 1 m are selected to test the positioning accuracy. In order to show the positioning error intuitively, an appropriate relative coordinate system is established so that the x and y coordinates of the 25 reference points are exact integers 1–5. As shown in Figure 7, in test area 1, the observation data of six pseudolites are collected in real-time by the receiver at an interval of 1 m, the collection time of each node is 30 s. The collected data is used as the input of the measured part of the neural network after normalization processing.

In the simulating calculation process, there are multiple propagation paths between the transmitting and receiving points. The direct path has the shortest transmission distance and the strongest reaching power. With the increase of the propagation distance and the number of reflection and transmission times, the signal transmission distance and the corresponding transmission delay also increase, and the pseudolite signal power attenuation is more serious until it is lower than the receiver signal power threshold. As shown in Figures 8 and 9, the electromagnetic information of 400 test points in text area 1 and 100 test points in text area 2 is obtained through 3D ray tracing calculation, which is converted and normalized as the simulation input part. Due to space limitations, Figures 8 and 9 show some examples of electromagnetic calculation results at 10 receiving test points.

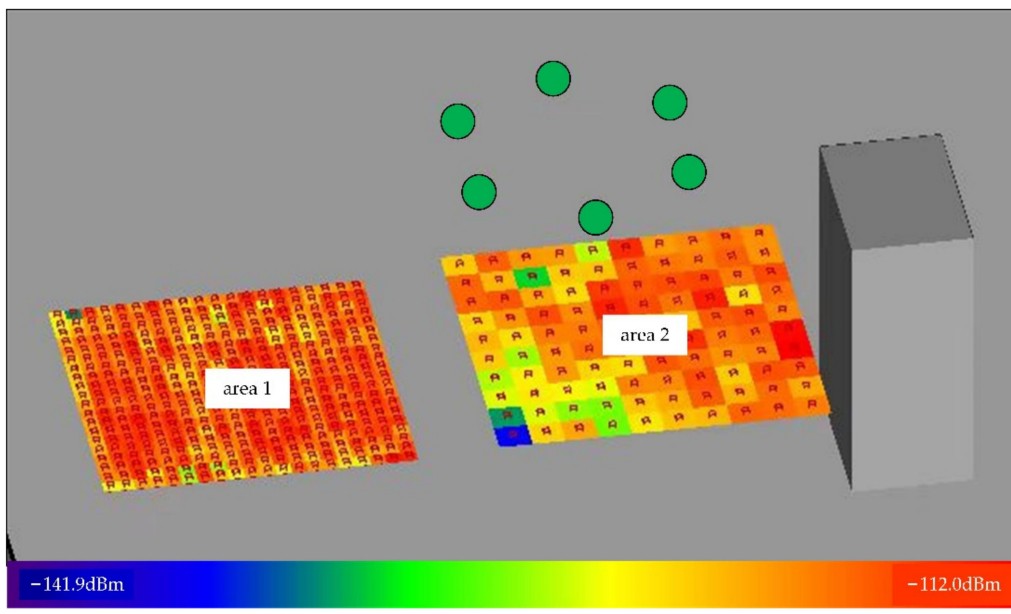

**Figure 8.** Simulation result of RSS.

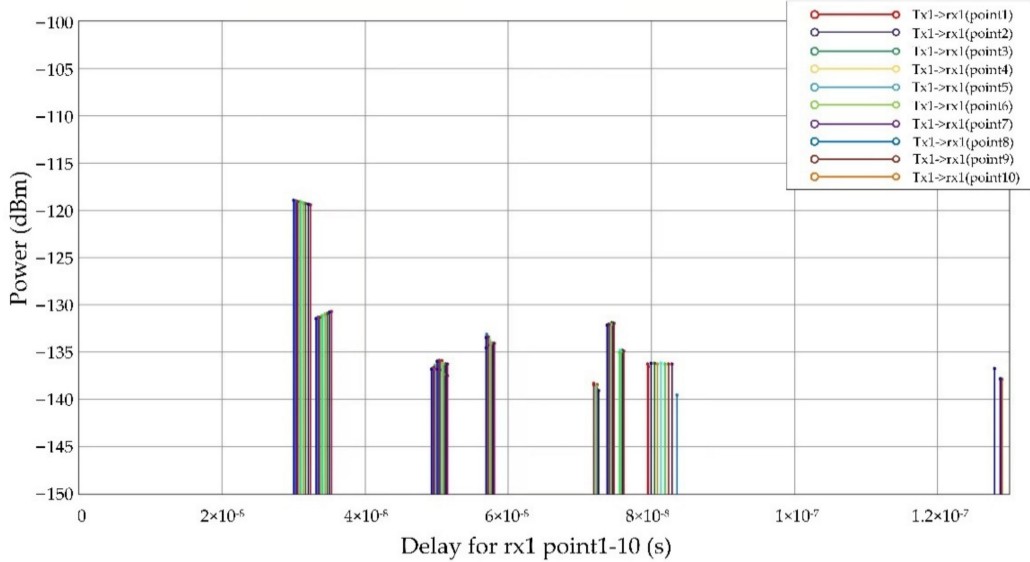

**Figure 9.** Simulation result of TOA.

### 5.3. The Location Result of the Area with Fingerprints Collection

The positioning data results of the two experiments are shown in Table 2. Figure 10 shows the test result of neural network positioning without simulation fingerprint data, the root mean square positioning error is 1.0696 m, Figure 11 shows the test result of neural network positioning with simulation fingerprint data, the root mean square positioning error is 0.4850 m. By comparing the results qualitatively and quantitatively, it can be seen that the improved artificial neural network adding deterministic simulation input features reduces the positioning test error (the distance between the real position and the positioning result is shortened), and the root means square error is reduced by 54.7%.

**Table 2.** Positioning results of collected fingerprint area.

| y | x | 1 | 2 | 3 | 4 | 5 |
|---|---|---|---|---|---|---|
| 1 | before | (1.9830, 0.2349) | (2.1571, 1.0887) | (1.8036, 0.2219) | (5.0260, 0.1302) | (4.0807, 0.2075) |
| | after | (0.9059, 1.2780) | (1.7449, 1.2948) | (3.2997, 1.1568) | (4.4700, 1.1110) | (4.8133, 1.3478) |
| 2 | before | (1.6671, 1.8230) | (2.2034, 2.6458) | (3.1488, 1.7992) | (5.0661, 1.8178) | (4.6179, 1.0020) |
| | after | (1.1044, 1.9267) | (1.8790, 1.5387) | (2.7085, 1.7319) | (4.0649, 1.8837) | (4.8270, 2.0046) |
| 3 | before | (1.6171, 2.0356) | (1.6278, 3.3335) | (5.0012, 3.5755) | (4.9678, 3.6350) | (3.8882, 3.2848) |
| | after | (0.5948, 2.780) | (1.7703, 4.000) | (3.3963, 3.1220) | (3.7177, 2.5306) | (5.3400, 2.5079) |
| 4 | before | (0.7725, 3.5277) | (2.4902, 4.9436) | (3.1220, 4.9431) | (4.2499, 4.0586) | (5.0921, 2.8164) |
| | after | (0.8360, 3.8335) | (1.7156, 4.3727) | (3.2457, 3.5751) | (5.0010, 4.3576) | (4.9925, 4.4191) |
| 5 | before | (1.4846, 5.6723) | (1.1861, 3.9457) | (3.4669, 3.8620) | (5.0229, 5.6108) | (5.5693, 5.5765) |
| | after | (0.6448, 4.8669) | (2.1337, 4.7858) | (3.0364, 5.4668) | (4.3620, 5.1035) | (4.5519, 4.9107) |

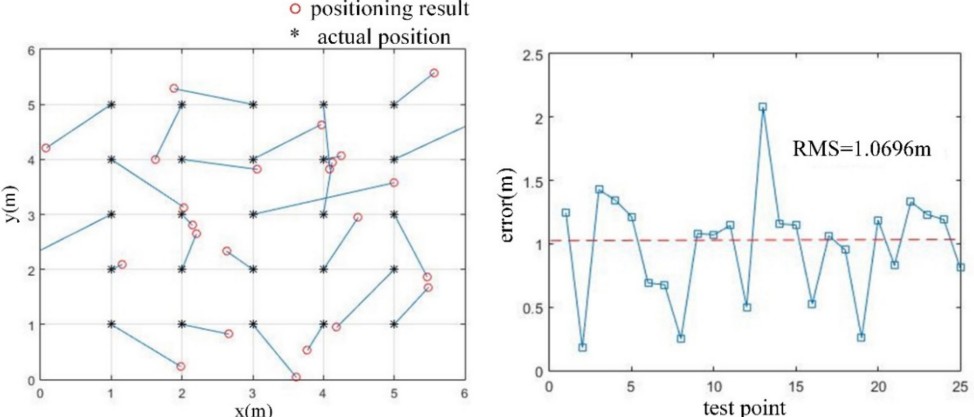

**Figure 10.** ANN test results without deterministic simulation input.

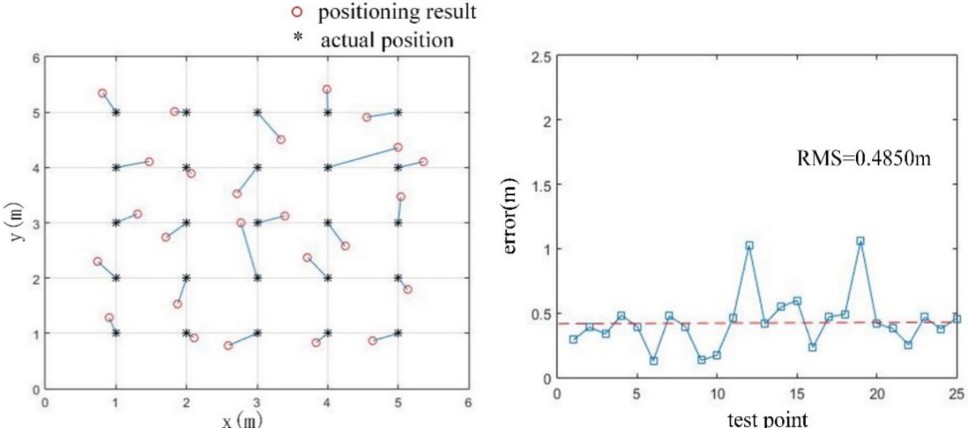

**Figure 11.** ANN test results with deterministic simulation input.

To compare the performance of different algorithms, we choose the commonly used KNN and SVM methods in fingerprint localization, and methods in [10,13] to compare with our method. The KNN and SVM model can be directly invoked through the deep learning framework Keras library. Similarly, the positioning calculation is performed at the 25 position reference points of test area 1 and compared with the true value points measured by the total station. Figure 12 is the positioning straight line distance error, and Figure 13 is the cumulative distribution function of the error. It can be seen that the root mean square (RMS) error of our method is 0.4850 m and the maximum error is 1.06 m, of which 92% are better than 1 m. Comparative experiments show that our method has higher localization accuracy than other methods, the detailed error analysis is shown in Table 3.

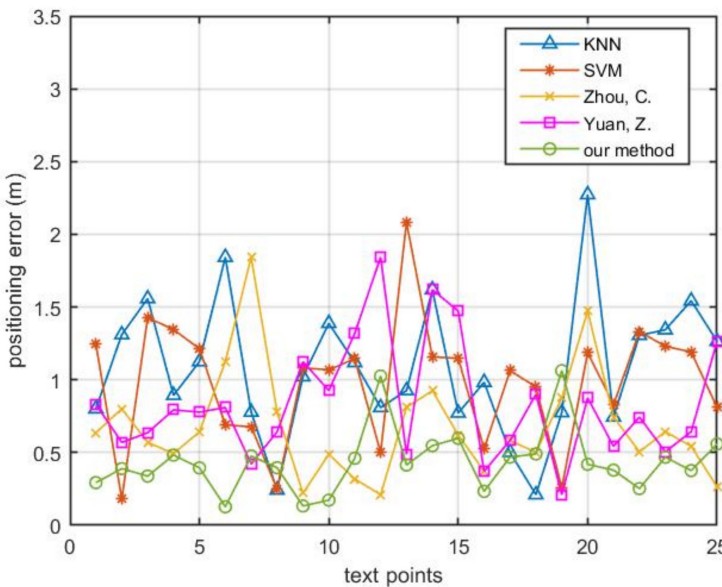

**Figure 12.** Comparison of positioning errors of different positioning algorithms.

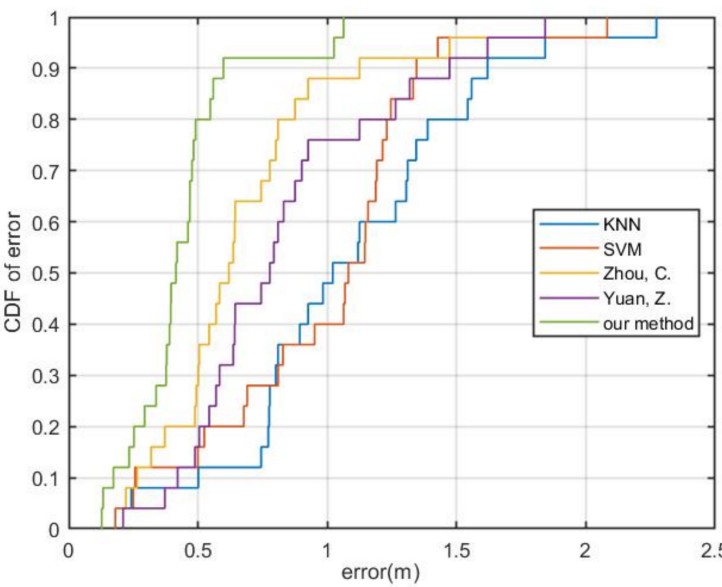

**Figure 13.** Cumulative Distribution Function of errors from different positioning algorithms.

**Table 3.** Comparison of Location Accuracy of different fingerprint positioning algorithms.

| Algorithm | KNN | SVM | Yuan, Z. [10] | Zhou, C. [13] | Our Method |
|---|---|---|---|---|---|
| RMS error (m) | 1.1821 | 1.0696 | 0.7714 | 0.9253 | 0.4850 |
| 95% error (m) | 2.2727 | 1.8424 | 1.6194 | 1.4727 | 1.026 |

*5.4. The Location Result of the Area without Fingerprints Collection*

The results of the positioning data of the experiment are shown in Table 4. Figure 14 is the location test result of test area 2, and the root mean square location error is 1.1237 m. The results show that the rough positioning can be completed by simply relying on the fingerprint positioning data generated by the simulation calculation, and it can also meet the need of low-precision positioning without manual fingerprints collection.

**Table 4.** Positioning results of uncollected fingerprint area.

| x<br>y | 1 | 2 | 3 | 4 | 5 |
|---|---|---|---|---|---|
| 1 | (1.2377, 0.2349) | (0.6923, 1.0887) | (1.6501, 0.2219) | (3.7950, 0.1302) | (5.6715, 1.9023) |
| 2 | (2.8339, 1.8230) | (1.5664, 2.6458) | (3.0349, 2.2396) | (3.8759, 3.0136) | (3.7925, 1.3151) |
| 3 | (0.0412, 2.4235) | (2.3426, 3.7335) | (3.7254, 2.4245) | (5.4897, 2.365) | (5.7172, 3.2848) |
| 4 | (1.8622, 3.5277) | (2.1784, 3.5315) | (2.9369, 3.7992) | (4.4090, 4.6586) | (5.6302, 6.1836) |
| 5 | (1.3188, 4.3277) | (2.7694, 6.0543) | (3.7147, 3.8620) | (5.4172, 5.6108) | (5.4889, 6.1659) |

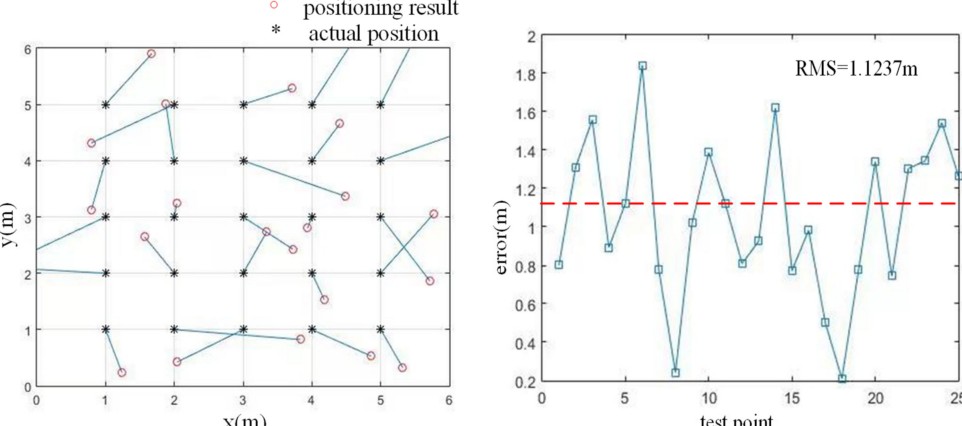

**Figure 14.** Positioning results of the area without collected fingerprints.

## 6. Conclusions

In this paper, an indoor pseudolite CSI fingerprinting method based on ray tracing and artificial neural network is proposed. According to the positioning requirements of large indoor scenes, the method uses the characteristic parameters generated by deterministic modeling to expand the input data set of an artificial neural network. The test environment is built in the arrival hall of an airport, and the positioning performance of the method has been tested and verified using the pseudolite positioning terminal developed by the State Key Laboratory of Satellite Navigation System and Equipment. The results show that the positioning accuracy is improved in the area where fingerprints have been actually collected, preliminary positioning can also be completed in the area without fingerprints collection.

Our current work is still rather preliminary at this stage, mostly due to the fact that the characteristics of simulated and measured CSI data are not sufficiently diverse. In future work, we intend to propose and analyze more CSI features such as AOA observations and Doppler observations. We also intend to apply this approach to dynamic localization testing to improve pedestrian localization accuracy in large venues.

**Author Contributions:** All authors contributed to the manuscript and discussed the results. All authors together developed the idea that led to this paper. Y.L. and J.L. conceived the experiments and analyzed the data. H.L. provided critical comments and contributed to the final revision of the paper. B.Y. contributed to the expression and the design of programs. Y.L. wrote the manuscript and all the authors participated in amending the manuscript. All authors have read and agreed to the published version of the manuscript.

**Funding:** This work was supported in part by the National Key Research and Development Plan of China (project: General and application verification of large-scale underground scalable PNT system: No. 2021YFB3900801) and Foundation of Technological Innovation Guidance Plan of Hebei (project: Research and application of key technologies of Beidou micro base station indoor hybrid positioning system for the Beijing Winter Olympics: No. 21477603D).

**Data Availability Statement:** Not applicable.

**Conflicts of Interest:** The authors declare no conflict of interest.

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
