# Peer review of "A CSI Fingerprint Method for Indoor Pseudolite Positioning Based on RT-ANN"

_futureinternet, doi:10.3390/fi14080235_

Round 1

Reviewer 1 Report

Kindly see the attached file for comments.

Reviewer 2 Report

This paper presents a neural network approach to indoor classification using channel state features.

+ The idea of using an artificial neural network and the general approach of using machine learning is interesting for the problem since the underlying system is so hard to model.

+ The selection of features adopted is appropriate and seems to contribute to different causes of inaccuracies in other techniques. 

+ The real-world experimental result shows that the work is promising.

The work can, however, be significantly improved. 

- Indoor localization is a widely studied area and is highly competitive. The authors must do a thorough job of comparing, both quantitatively and qualitatively to other works. 

- The neural network presented in this work will be able to model a particular location with all its physical obstructions. However, it seems that the authors do not offer a lot of detail about how generalizable the model is. For example, how much additional data collection and training/retraining will be necessary if a new furniture set is added to the setting. 

- The paper can also benefit from thorough proofreading to improve typographical errors. Additionally, some terms are used without definition in the paper. For example, CSI is expanded later in the paper, but the first few instances appear without an expansion. Besides, more importantly, the loss functions are stated for the first time in algorithm 1. 

Overall, the work is exciting and the real-world experiment is a huge strength of the paper. 

Reviewer 3 Report

Dear Authors, 

Please find the attached file for your reference. Please update the paper based on the comments and resubmit it. 

Best Regards 

Round 2

Reviewer 1 Report

All comments are resolved except for performance comparison. The KNN and SVM are not state-of-the-art approaches. Authors are strongly advised to select already proposed positioning approaches and add a performance comparison Table regarding the positioning accuracy.

Reviewer 2 Report

The authors have made significant improvements to the paper and addressed all reviewer comments. 

Author Response

We appreciated for reviewers’ warm work earnestly!

Reviewer 3 Report

Dear Authors,

Thank you for addressing all my concerns, and I don't have any further concerns on the paper. 

Best regards

Author Response

(The authors gave the same response as above.)
